# The Interactive Relationship between Street Centrality and Land Use Intensity—A Case Study of Jinan, China

**DOI:** 10.3390/ijerph20065127

**Published:** 2023-03-14

**Authors:** Chengzhen Song, Qingfang Liu, Jinping Song, Ding Yang, Zhengyun Jiang, Wei Ma, Fuchang Niu, Jinmeng Song

**Affiliations:** 1Faculty of Geographical Science, Beijing Normal University, Beijing 100875, China; 2Institute of Geographic Sciences and Natural Resources Research, Chinese Academy of Sciences, Beijing 100101, China

**Keywords:** street centrality, land use intensity, POI data, multiple centrality assessment model, Jinan, China

## Abstract

It is of great significance to study the interactive relationship between urban transportation and land use for promoting the healthy and sustainable development of cities. Taking Jinan, China, as an example, this study explored the interactive relationship between street centrality (SC) and land use intensity (LUI) in the main urban area of Jinan by using the spatial three-stage least squares method. The results showed that the closeness centrality showed an obvious “core-edge” pattern, which gradually decreased from the central urban area to the edge area. Both the betweenness centrality and the straightness centrality showed a multi-center structure. The commercial land intensity (CLUI) showed the characteristics of multi-core spatial distribution, while the residential land intensity (RLUI) and public service land intensity (PLUI) showed the characteristics of spatial distribution with the coexistence of large and small cores. There was an interactive relationship between SC and LUI. The closeness centrality and straightness centrality had positive effects on LUI, and LUI had a positive effect on closeness centrality and straightness centrality. The betweenness centrality had a negative impact on LUI, and LUI also had a negative impact on betweenness centrality. Moreover, good location factors and good traffic conditions were conducive to improving the closeness and straightness centrality of the regional traffic network. Good location factors, good traffic conditions and high population density were conducive to improving regional LUI.

## 1. Introduction

Urban transportation and land use have always been popular research fields in urban geography, urban planning and urban economics. Relevant research can be traced back to the model of urban internal regional structure proposed by the “Chicago School” in the early 20th century (Park, 1915), such as the concentric circle model proposed by Burgess in 1925 [1], the fan model proposed by Hoyt in 1939 [2] and the multi-core model proposed by Harris and Ullman in 1945 [3]. All of these models mentioned the importance of transportation in shaping urban spatial structure. Moreover, the classic Western economic model also emphasized the importance of urban transportation in shaping urban spatial structure. For example, in the 1960s, Alonso [4], an American land economist, proposed the rent competition theory, and Mills [5] (1972) and Muth [6] (1969) proposed the single-center model.

In China, since the reform and opening up of the country, the process of urbanization has been rapidly advancing. From 1978 to 2019, China’s urbanization rate increased rapidly from 17.9% to 60.6%. In the context of rapid urbanization, China’s urban space has expanded rapidly, and many large- and medium-sized cities have emerged in the process of development in a “big pie” type of spatial disorder development mode, which has triggered a series of urban problems, such as overcrowding in urban central areas, a waste of land resources around the city and a decline in the quality of the urban environment [7]. In addition, in the process of the construction of new urban areas, problems such as excessive construction space of new urban areas and excessive motorization of roads generally exist, which are manifested in the excessive sizes of street space and the low density of road networks [8,9]. The establishment of a reasonable, complete and efficient urban transportation system is conducive to alleviating urban traffic congestion, facilitating citizens’ travel, optimizing the urban internal spatial structure and improving land use efficiency, which is of great significance to improving the quality of life for urban residents and promoting sustainable urban development.

As an important part of urban infrastructure, the transportation system plays a key role in urban land use planning and development. Moreover, the spatial structure of urban land use also guides the planning and construction of urban transportation infrastructure. Therefore, urban traffic and land use interact with each other [10]. However, current scholars mainly carry out one-way impact studies and linear correlation studies on urban traffic and land use, and there are few studies on the interaction between urban traffic and land use.

According to the 2017 Transportation Analysis Report of China’s Major Cities, Jinan once again ranked first in the list of China’s congested cities with a peak congestion delay index of 2.067. In 2018 and 2019, Jinan ranked among the top 10 most congested cities in China. Traffic congestion seriously restricts the healthy and sustainable development of cities. Different from the urban development mode of the United States, the central area of Chinese cities is the agglomeration area of production and the life of residents, as well as the concentration area of urban transportation and land development. How to coordinate the relationship between urban traffic and land use, further optimize the urban spatial structure and promote the healthy and high-quality development of the city, which are important parts of the future urban development of Jinan.

This paper took Jinan, a typical city, as an example, and selected the spatial three-stage least squares method to explore the interactive relationship between SC and LUI. Moreover, this paper explored the factors that influence SC and LUI. Compared with existing studies, this paper has the following advantages: First, this study makes up for the deficiencies in the current quantitative research on the interactive relationship between urban transportation and land use. Second, this study took the concentrated area of urban transportation and land development in China (the main urban area of Jinan) as the research object, as it is representative and valuable.

The research structure of this paper is as follows: Section 2 is the literature review. Section 3 comprises the study area overview, methods and data sources. Section 4 is the research results. Section 5 is the discussion of the research results. Section 6 is the main conclusions.

## 2. Literature Review

At present, the academic community has accumulated some research experience in urban transportation and land use. Relevant research mainly focuses on the following aspects:

### 2.1. Exploring the Impact of Urban Traffic on Land Use

European and American scholars have carried out earlier studies on the impact of urban traffic on land use, and they have improved urban land use patterns by optimizing transportation facilities. For example, Schaeffer explored the relationship between urban traffic development and urban spatial form, and analyzed the importance of transportation in shaping urban spatial form [11]. Giannopoulos systematically analyzed the important impact of transportation technology innovation on urban morphology from the perspective of technological innovation [12]. Moon explored the influence of Washington metro stations on commercial land and residential land, and found that traffic stations could promote the outward diffusion of the population in the central business district [13]. Baerwald found that traffic accessibility was the key factor to determining urban land for residential land development by exploring the impact of traffic accessibility on residential land development [14]. Rui et al. took Stockholm as the research object and found that road network centrality could affect the spatial distribution of land use types [15]. With the rapid development of urbanization in China, many Chinese scholars began to pay attention to the impact of urban traffic on land use. For example, taking Guangzhou as the research object, Mao Jiangxing explored the impact of the urban transportation system on land use price by using spatial analysis technology and the multivariate statistical analysis method [16]. Sun Jiuwen believed that the construction of urban rail transit could change the urban spatial development pattern, which was of great significance for alleviating urban traffic congestion and achieving high-quality urban development [17].

### 2.2. Exploring the Impact of Land Use on Urban Traffic

Some scholars found that urban land use structure could guide the rational planning and construction of urban transportation facilities. For example, Giuliaono found that urban land development could promote the development of urban transportation infrastructure after reaching a certain level [18]. Johnson explored the impact of land use on the demand for public transport infrastructure and concluded that improving the mix of urban land use could improve the demand for public transport infrastructure [19]. Based on the survey data of Boston and Hong Kong, Zhang Ming concluded through model empirical analysis that urban land use status had an important impact on citizens’ choice of travel mode [20]. Ding Chengri made a comparative study on the influence of the spatial structure of single-center and multi-center cities on the urban transportation system, and found that the single-center structure was conducive to the development of urban public transportation, while the multi-center urban structure did not necessarily reduce the demand for urban transportation, and the impact of mixed urban land use on urban transportation was uncertain [21].

### 2.3. Exploring the Relationship between Urban Traffic and Land Use

Since the 1960s, some developed areas, such as Europe and the United States, have begun to carry out some studies on urban development, land use and transportation planning by creating an integrated model of transportation planning and land use, such as the Lowry model, methods based on mathematical planning, input–output methods, and rent-competition function methods based on urban economics [22,23,24]. In recent years, some Chinese scholars have also begun to pay attention to research on urban transportation and land use. For example, Wang Shuai took Shenzhen as an example and explored the correlation between road network centrality and land use intensity using the Pearson correlation coefficient [25]. Taking the main urban area of Chongqing as the research object, Tian et al. measured the coordination degree of land use and urban traffic by using the DEA method [26]. Taking Hubei Province as an example, Yin Chaohui explored the relationship between the centrality of the road network and the landscape pattern of land use by using the Spearman correlation analysis method [27]. Yin Guanwen took Jinan as an example to explore the correlation and heterogeneity between the centrality of the urban road network and land use intensity [28].

### 2.4. Exploring the Urban Traffic Network

The rapid development of cyberspace science has greatly promoted the relevant research into the urban transportation network, which has provided convenient technical support for exploring the relationship between the urban transportation network and land use [29]. At present, there are two popular models for studying the traffic network: the spatial syntax model [30,31,32] and the multiple centrality assessment model [33,34,35]. In the 1980s, the British scholar Bill Hillier first proposed the spatial syntax model [36], which cannot only effectively measure the local spatial accessibility of a region, but also measure the overall spatial relevance of a region. It is often used to explore the correlation between urban economic activities and road network centrality [37]. In 2006, Cruciti proposed a multiple centrality assessment model based on the spatial syntax theory, which reflects the importance of road network nodes by measuring the centrality of road nodes [38,39]. Owing to the multiple centrality assessment model which measures the actual distance, the calculation results are more scientific and reliable [40].

In general, current academic circles mainly focus on one-way impact research on urban traffic and land use, such as the impact of urban traffic on land use or the impact of land use on urban traffic [41,42]. In the correlation study of urban traffic and land use, the Pearson correlation coefficient was mainly used to study the correlation between the two [43,44]. However, there is little research on the interactive relationship between urban traffic and land use, and the existing research mainly focuses on qualitative analysis. Moreover, in the research into the urban traffic network, the spatial syntax and multiple centrality assessment models are mainly used for quantitative evaluation.

## 3. Study Area, Methods and Data Sources

### 3.1. Study Area

Jinan, the capital of Shandong Province, is also a city famous for spring water. A large number of karst caves are distributed underground in Jinan, which causes great difficulties for urban subway construction and leads to the lagging development of local rail transit. In addition, restricted by the topographical conditions, the Jinan urban area presents a narrow and long belt space form from east to west and short from north to south [45]. This often leads to serious traffic congestion in Jinan. This paper took Jinan, China, as the research case, which was typical and representative. With reference to the Master Plan of Jinan City (2011–2020) and the distribution of the urban road network in Jinan, the study area determined by this study is shown in Figure 1.

### 3.2. Methods

#### 3.2.1. Multiple Centrality Assessment Model

The multiple centrality assessment model takes the urban road as the edge of the network and the intersection point of the road and the end point of the road as the network node, and calculates the centrality of the urban traffic network by calculating the distance between actual road nodes [46,47]. The model mainly evaluates the importance of the road network by calculating closeness centrality, betweenness centrality and straightness centrality.

Closeness centrality (*C^C^*) represents the degree of closeness between a transport network node and all other nodes in the transport network, reflecting the reachability of this node in the network. The formula for calculating the closeness centrality of a road node *i* is
(1)$$C_{i}^{C}=\frac{N-1}{\sum_{j=1;j\neq i}^{N}d_{ij}}$$
where *N* is the number of nodes in the traffic network and *d_ij_* is the shortest distance between node *i* and node *j*.

Betweenness centrality (*C^B^*) reflects the transfer capacity of road nodes in the traffic network. The stronger the betweenness centrality of road nodes is, the stronger the hub role it plays in the traffic network is. The formula for calculating the betweenness centrality of the road node *i* is
(2)$$C_{i}^{B}=\frac{1}{\left(N-1\right)\left(N-2\right)}\sum_{j=1;k=1;j\neq k\neq 1}^{N}\frac{n_{jk}\left(i\right)}{n_{jk}}$$
where *N* is the number of nodes in the traffic network, *n_jk_* is the number of shortest paths between node *j* and node *k* and *n_jk_*(*i*) is the number of paths through node *i* in the shortest paths between node *j* and node *k*.

Straightness centrality (*C^S^*) reflects the importance of road nodes. The stronger the straightness centrality, the stronger the traffic efficiency of the traffic network is.
(3)$$C_{i}^{S}=\frac{1}{N-1}\sum_{j=1;j\neq i}^{N}\frac{d_{ij}^{Eucl}}{d_{ij}}$$
where *N* is the number of nodes in the traffic network, *d_ij_^Eucl^* represents the Euclidean distance between node *i* and node *j* and *d_ij_* is the shortest distance between node *i* and node *j*. 

#### 3.2.2. Kernel Density Estimation (KDE)

KDE calculates the density of points distributed in a specific window, and takes the sum of the density of all points in the window as the kernel density value of the grid center [48,49]. In this paper, KDE was used to smooth the POI data of the main urban area of Jinan, and a continuous spatial distribution map was obtained to indirectly represent the LUI.
(4)$$\hat{f}\left(x\right)=\frac{1}{nh^{d}}\sum_{i=1}^{n}K\left(\frac{x-x_{i}}{h}\right)$$
where *K* represents the kernel function, *h* is the threshold, *n* is the number of points within the threshold range and *d* represents the data dimension. In this study, after considering the smoothness of the data and the detail of the reflected data, the bandwidth selected was 1000 m.

#### 3.2.3. Spatial Three-stage Least Squares Method

The spatial three-stage least squares method constructs a model using simultaneous equations for spatial econometric analysis. This method not only considers the potential spatial correlation of endogenous variables, but also considers the correlation between the random error terms of each equation, thus making the results more scientific and effective. It cannot only avoid the traditional simultaneous equation model and ignore the possible spatial spillover effect between variables, but it can also solve the problem of variable endogeneity that may be generated by the spatial econometric model [50].

The spatial correlation test found that the global Moran index of SC and LUI in the main urban area of Jinan was higher than 0.8, and passed the significance test, indicating that there was significant spatial autocorrelation between the two. Therefore, spatial factors need to be taken into account in the analysis of influencing factors. In this study, the spatial three-stage least squares method was used to explore the interactive relationship between SC and LUI. The established spatial simultaneous equation was as follows:$$SC_{i}=\alpha_{0}+\alpha_{1}\sum_{j\neq i}^{n}W_{ij}SC_{j}+\alpha_{2}\sum_{j\neq i}^{n}W_{ij}LUI_{j}+\alpha_{3}LUI_{j}+\alpha X_{i}+\varepsilon_{i}$$
(5)$$LUI_{i}=\beta_{0}+\beta_{1}\sum_{j\neq i}^{n}W_{ij}LUI_{j}+\beta_{2}\sum_{j\neq i}^{n}W_{ij}SC_{j}+\beta_{3}SC_{j}+\beta Z_{i}+\eta_{i}$$
where *i* represents the sample region; *SC_i_* and *LUI_i_* represent SC and LUI, respectively; *W_ij_* represents the spatial weight matrix; *X_i_* and *Z_i_* represent a group of control variables that affect the SC and the LUI; *ε_i_* and *η_i_* represent unobservable factors; *α*_1_ represents the spatial spillover estimation coefficient of the SC between neighboring regions; *β*_1_ represents the spatial spillover estimation coefficient of the LUI between neighboring regions; *α*_2_ and *β*_2_ represent the spatial interaction used to test the SC and the LUI and *α*_3_ and *β*_3_ are used to characterize the endogenous relationship between the SC and the LUI.

Endogenous variables: Street centrality (SC) and land use intensity (LUI). Considering the availability, scientificity and completeness of data, and referring to the existing research results [51,52,53,54], the control variables *X_i_* that affect the SC were selected. Road network density (ND): the control variable was represented by the road length within 1 km^2^. Economic factors (EC): the control variable was represented by the regional GDP within 1 km^2^, unit: 10,000 yuan/km^2^. Location factor (LOC): the control variable was characterized by the distance to the CBD. 

The control variables *Z_i_* that affect the LUI: Population factor (POP): the control variable was represented by the population within 1 km^2^, unit: person/km^2^. Economic factors (EC): the control variable was represented by the regional GDP within 1 km^2^, unit: 10,000 yuan/km^2^. Traffic factor (TRA): the control variable was measured by bus stop density, which was obtained by calculating the number of bus stops within a 1 km^2^ grid. Location factor (LOC): the control variable was characterized by the distance from the CBD.

Considering the influence of variable regional units, this paper constructed spatial grid units for quantitative analysis. First, the fishnet creation tool in ArcGIS software was used to construct a square grid covering the study area with a side length of 1 km. Then, the grid was used as a statistical unit to calculate the SC, LUI and other control variable indexes through spatial correlation and other methods.

First of all, in order to avoid the influence of heteroscedasticity and multicollinearity on the research results, the original data of variables were processed logically and tested using the variance inflation factor (VIF). The test results showed that the VIF of each variable was less than 7.5, so it could be considered that there was no multicollinearity between variables. Secondly, the test results of the validity and feasibility of the model showed that the R^2^ of the SC equation and of the LUI equation were both greater than 0.5, indicating that the fitting effect of the model results was good. Therefore, the model was valid and feasible. 

### 3.3. Data Sources and Processing

The traffic network data of Jinan were obtained from the China National Geographic Information Public Service Platform. POI data can record the attribute information and spatial location of buildings or geographical entities. POI data of Jinan in 2020 were collected based on Gaode API.

The ArcGIS software was used to extract the highways, main roads, secondary roads and main branches of the Jinan main urban area. Secondly, disordered and disconnected routes were eliminated, and repeated routes were screened and combined with remote sensing satellite images. The traffic network data of the study area after processing are shown in Figure 2.

The POI data can intuitively reflect the spatial distribution of various urban functional elements and facilities. This paper used the re-classified POI data kernel density smoothing results to indirectly represent the LUI. Firstly, data cleansing was performed on the obtained POI data. Then, the POI data of commercial facilities, residential facilities and public service facilities were selected for research (Table 1).

## 4. Results

### 4.1. Spatial Characteristics of SC and LUI

#### 4.1.1. Spatial Distribution Characteristics of SC

The closeness centrality showed an obvious “core-edge” pattern in space (Figure 3), with higher closeness centrality in the central area and lower closeness centrality in the marginal area. The closeness centrality gradually decreases from the urban central area to the marginal area. Specifically, high-value areas were mainly concentrated in the areas around Daming Lake, such as CBD, the commercial port area, etc. This area is the central area of the main urban area of Jinan, with a large number of commercial and tourist facilities, and the traffic network in this area is densely distributed. The shortest distance from road nodes to other nodes in the traffic network is small, and the traffic network accessibility is strong.

The spatial distribution of betweenness centrality was different from that of closeness centrality. High-value areas were mainly distributed in the main urban traffic roads, such as Quancheng Road, Luoyuan Street, Jiefang Road, Beiyuan Street, Jingshi Road and Lishan Road. The commercial facilities, tourist facilities and public service facilities on both sides of these main traffic roads are densely distributed, and the daily traffic flow and vehicle flow are large, which bear most of the daily traffic flow in the main urban area of Jinan.

The straightness centrality showed an obvious multi-center structure in space. There were three high-value clusters in the east, middle and west, such as the Lashan Business District in the west, the West Railway Station Business District in the middle and the High-Tech Development Zone in the east. These areas are far from the old urban area of Jinan, and the roads are mainly large traffic trunk roads, such as Gongye North Road, Gongye South Road, Jingshi East Road, Tourist Road and Jingshi West Road. The straightness centrality of road nodes in these regions was high, indicating that the path distance between two road nodes in this region is closer to the linear distance between them and that the regional traffic efficiency is high.

#### 4.1.2. Spatial Distribution Characteristics of LUI

In this study, reclassified POI data of commercial facilities, residential facilities and public service facilities were used to indirectly characterize the following three types of land use: commercial land, residential land and public service land, and LUI was obtained through kernel density smoothing (Figure 4).

CLUI showed a multi-core distribution pattern in space. CBD was the largest agglomeration center, and several secondary agglomeration centers were distributed around it. High-value areas of CLUI were mainly distributed around Daming Lake and the CBD area within the second ring road. This area is the old urban area of Jinan, where many large commercial service facilities are distributed, such as Ginza Shopping Plaza, World Trade Plaza and China Merchants Building.

RLUI showed the spatial distribution characteristics of the coexistence of large and small cores. High-value RLUI areas were mainly distributed in the eastern area of Daming Lake and the area around CBD, forming a large core. Residential communities in this area are densely distributed, such as Qinghou Community, Zhengjuesi Community, Shunyu Community and Jinjiling Villa District. There were two small cores in the east and southwest of the large core of RLUI.

The distribution of PLUI was similar to that of RLUI, which also showed the spatial distribution characteristics of coexisting large and small cores. High-value areas of PLUI were mainly concentrated within the second ring road, forming a large core. A large number of public service facilities are distributed in the area, such as Shandong Museum, Shandong Radio and Television News, Qilu Hospital of Shandong University and Qianfo Mountain Hospital of Shandong Province. There were two small cores in the east and southwest of the large core of PLUI.

### 4.2. Interactive Relationship between SC and LUI

#### 4.2.1. Interaction between SC and CLUI

The estimated results of the closeness centrality equation (Model 1) are shown in Table 2. The coefficient of the lag term (W × lnLUI) of CLUI was positive and passed the significance test, indicating that the CLUI in the adjacent area could promote an improvement in the closeness centrality of the local area. The coefficient of CLUI (lnLUI) was positive and passed the significance test, indicating that the increase in CLUI could promote an improvement in the regional closeness centrality. The coefficient of the lag term (W × lnSC) of closeness centrality was positive and passed the significance test, indicating that improving the closeness centrality in the adjacent area could promote the local closeness centrality. The influence coefficient of the road network density (lnND) was positive and passed the significance test, indicating that the greater the road network density, the higher the closeness centrality of road nodes. The influence coefficient of the location factor (lnLOC) was negative and passed the significance test, indicating that the farther away from the city center, the smaller the closeness centrality of road nodes.

The estimated results of the CLUI equation (Model 2) are shown in Table 2. The coefficient of the lag term (W × lnSC) of closeness centrality was positive and passed the significance test, indicating that an improvement in closeness centrality in the adjacent area could promote an improvement in local CLUI. The influence coefficient of closeness centrality (lnSC) was positive and passed the significance test, indicating that improving the closeness centrality was conducive to improving CLUI. The influence coefficient of the economic factors (lnEC) was positive and passed the significance test, indicating that improving the level of economic development was conducive to improving CLUI. The influence coefficient of the location factor (lnLOC) was negative and passed the significance test, indicating that the farther away from the city center, the lower the CLUI. The influence coefficient of population size (lnPOP) was positive and passed the significance test, indicating that the higher the local population density, the higher the CLUI. The influence coefficient of the traffic factor (lnTRA) was positive and passed the significance test, indicating that the higher the density of public transport stations, the higher the CLUI.

The estimated results of the betweenness centrality equation (Model 3) are shown in Table 2. The coefficient of the lag term (W × lnLUI) of CLUI was positive and passed the significance test, indicating that improving the CLUI in the adjacent areas could promote an improvement in the betweenness centrality of the local area. The coefficient of the lag term (W × lnSC) of betweenness centrality was negative and passed the significance test, indicating that improving the betweenness centrality in the adjacent region could lead to a decrease in the local betweenness centrality. The influence coefficient of the road network density (W × lnND) was negative and passed the significance test, indicating that the higher the road network density, the lower the betweenness centrality.

The estimated results of the CLUI equation (Model 4) are shown in Table 2. The coefficient of the lag term (W × lnSC) of betweenness centrality was positive and passed the significance test, indicating that improving the betweenness centrality in the adjacent areas could promote an improvement in local CLUI. The influence coefficient of the location factor (lnLOC) was negative and passed the significance test, indicating that the farther away from the city center, the lower the CLUI. The influence coefficient of population size (lnPOP) was positive and passed the significance test, indicating that the higher the population density, the higher the CLUI. The influence coefficient of the traffic factor (lnTRA) was positive and passed the significance test, indicating that increasing the density of public transport stations was conducive to improving the CLUI.

The estimated results of the straightness centrality equation (Model 5) are shown in Table 2. The coefficient of the lag term (W × lnLUI) of CLUI was positive and passed the significance test, indicating that the CLUI in the adjacent areas could promote an improvement in the straightness centrality of the local area. The influence coefficient of CLUI (lnLUI) was positive and passed the significance test, indicating that improving the CLUI could promote an improvement in the straightness centrality. The coefficient of the lag term (W × lnSC) of the straightness centrality was negative and passed the significance test, indicating that an increase in straightness centrality in the adjacent areas could lead to a decrease in local straightness centrality. The influence coefficient of the location factor (lnLOC) was negative and passed the significance test, indicating that the farther away from the city center, the lower the straightness centrality of road nodes.

The estimated results of the CLUI equation (Model 6) are shown in Table 2. The coefficient of the lag term (W × lnSC) of the straightness centrality was positive and passed the significance test, indicating that improving the straightness centrality in the adjacent areas could promote an improvement in local CLUI. The influence coefficient of the straightness centrality (lnSC) was positive and passed the significance test, indicating that an improvement in straightness centrality could promote an improvement in CLUI. The influence coefficient of the economic factors (lnEC) was positive and passed the significance test, indicating that improving the level of economic development could promote an improvement in CLUI. The influence coefficient of the location factor (lnLOC) was negative and passed the significance test, indicating that the farther away from the city center, the lower the CLUI. The influence coefficient of population size (lnPOP) was positive and passed the significance test, indicating that the higher the population density, the higher the CLUI.

#### 4.2.2. Interaction between SC and RLUI

The estimated results of the closeness centrality equation (Model 1) are shown in Table 3. The coefficient of the lag term (W × lnLUI) of RLUI was positive and passed the significance test, indicating that the RLUI in the adjacent area could promote an improvement in the closeness centrality of the local area. The coefficient of RLUI (lnLUI) was positive and passed the significance test, indicating that an increase in RLUI could promote an improvement in the closeness centrality. The influence coefficient of the road network density (lnND) was positive and passed the significance test, indicating that increasing the road network density could promote an improvement in closeness centrality. The influence coefficient of the economic factors (lnEC) was positive and passed the significance test, indicating that improving the level of economic development was conducive to improving the closeness centrality of road nodes. The influence coefficient of the location factor (lnLOC) was negative and passed the significance test, indicating that the farther away from the city center, the smaller the closeness centrality of road nodes.

The estimated results of the RLUI equation (Model 2) are shown in Table 3. The coefficient of the lag term (W × lnSC) of closeness centrality was positive and passed the significance test, indicating that an improvement in closeness centrality in the adjacent area could promote an improvement in local RLUI. The influence coefficient of closeness centrality (lnSC) was positive and passed the significance test, indicating that improving closeness centrality was conducive to improving RLUI. The influence coefficient of population size (lnPOP) was positive and passed the significance test, indicating that the higher the population density, the higher the RLUI. The influence coefficient of the traffic factor (lnTRA) was positive and passed the significance test, indicating that the higher the density of public transport stations, the higher the RLUI.

The estimated results of the betweenness centrality equation (Model 3) are shown in Table 3. The coefficient of the lag term (W × lnLUI) of RLUI was positive and passed the significance test, indicating that improving the RLUI in the adjacent areas could promote an improvement in the betweenness centrality of the local area. The influence coefficient of RLUI (lnLUI) was negative and passed the significance test, indicating that increasing the RLUI could lead to a decrease in betweenness centrality. The coefficient of the lag term (W × lnSC) of betweenness centrality was negative and passed the significance test, indicating that improving the betweenness centrality in the adjacent region could lead to a decrease in the local betweenness centrality. The influence coefficient of the road network density (W × lnND) was negative and passed the significance test, indicating that the higher the road network density, the lower the betweenness centrality.

The estimated results of the RLUI equation (Model 4) are shown in Table 3. The influence coefficient of betweenness centrality (lnSC) was negative and passed the significance test, indicating that an increase in betweenness centrality could lead to a decrease in RLUI. The influence coefficient of the location factor (lnLOC) was negative and passed the significance test, indicating that the farther away from the city center, the lower the RLUI. The influence coefficient of population size (lnPOP) was positive and passed the significance test, indicating that the higher the population density, the higher the RLUI.

The estimated results of the straightness centrality equation (Model 5) are shown in Table 3. The coefficient of the lag term (W × lnLUI) of RLUI was positive and passed the significance test, indicating that the RLUI in the adjacent areas could promote an improvement in straightness centrality of the local area. The influence coefficient of RLUI (lnLUI) was positive and passed the significance test, indicating that an improvement in RLUI was conducive to an improvement in straightness centrality. The coefficient of the lag term (W × lnSC) of the straightness centrality was negative and passed the significance test, indicating that an increase in straightness centrality in the adjacent areas could lead to a decrease in local straightness centrality. The influence coefficient of the location factor (lnLOC) was negative and passed the significance test, indicating that the farther away from the city center, the lower the straightness centrality of road nodes.

The estimated results of the RLUI equation (Model 6) are shown in Table 3. The influence coefficient of straightness centrality (lnSC) was positive and passed the significance test, indicating that an improvement in straightness centrality was conducive to an improvement in RLUI. The coefficient of the lag term (W × lnLUI) of the RLUI lag was positive and passed the significance test, indicating that improving RLUI in the adjacent areas could promote an improvement in local RLUI. The influence coefficient of the economic factors (lnEC) was positive and passed the significance test, indicating that an improvement in the economic development level was conducive to an improvement in RLUI. The influence coefficient of the location factor (lnLOC) was negative and passed the significance test, indicating that the farther away from the city center, the lower the RLUI. The influence coefficient of population size (lnPOP) was positive and passed the significance test, indicating that the higher the population density, the higher the RLUI.

#### 4.2.3. Interaction between SC and PLUI

The estimated results of the closeness centrality equation (Model 1) are shown in Table 4. The coefficient of the lag term (W × lnLUI) of PLUI was positive and passed the significance test, indicating that the PLUI in the adjacent area could promote an improvement in the closeness centrality of the local area. The coefficient of PLUI (lnLUI) was positive and passed the significance test, indicating that increasing the PLUI was conducive to promoting an improvement in closeness centrality. The influence coefficient of the road network density (lnND) was positive and passed the significance test, indicating that increasing road network density was conducive to promoting an improvement in closeness centrality. The influence coefficient of the economic factors (lnEC) was positive and passed the significance test at the level of 10%, indicating that an improvement in the economic development level was conducive to an improvement in closeness centrality. The influence coefficient of the location factor (lnLOC) was negative and passed the significance test, indicating that the farther away from the city center, the smaller the closeness centrality.

The estimated results of the PLUI equation (Model 2) are shown in Table 4. The coefficient of the lag term (W × lnSC) of closeness centrality was positive and passed the significance test, indicating that an improvement in closeness centrality in the adjacent area could promote an improvement in local PLUI. The influence coefficient of closeness centrality (lnSC) was positive and passed the significance test, indicating that an improvement in closeness centrality was conducive to an improvement in PLUI. The influence coefficient of the location factor (lnLOC) was negative and passed the significance test, indicating that the closer to the city center, the higher the PLUI. The influence coefficient of population size (lnPOP) was positive and passed the significance test, indicating that the higher the local population density, the higher the PLUI.

The estimated results of the betweenness centrality equation (Model 3) are shown in Table 4. The influence coefficient of PLUI (lnLUI) was negative and passed the significance test, indicating that an increase in PLUI could lead to a decrease in local betweenness centrality. The coefficient of the lag term (W × lnSC) of betweenness centrality was negative and passed the significance test, indicating that improving the betweenness centrality in the adjacent region could lead to a decrease in the local betweenness centrality. The influence coefficient of the road network density (W × lnND) was negative and passed the significance test, indicating that the higher the road network density, the lower the betweenness centrality. 

The estimated results of the PLUI equation (Model 4) are shown in Table 4. The influence coefficient of betweenness centrality (lnSC) was negative and passed the significance test, indicating that an increase in betweenness centrality could lead to a decrease in the PLUI. The coefficient of the lag term (W × lnLUI) of public service land was positive and passed the significance test, indicating that an improvement in the PLUI in the adjacent areas could promote an improvement in the local PLUI. The influence coefficient of the economic factors (lnEC) was positive and passed the significance test, indicating that improving the level of economic development could promote an improvement in the PLUI. The influence coefficient of the location factor (lnLOC) was negative and passed the significance test, indicating that the farther away from the city center, the lower the PLUI. The influence coefficient of population size (lnPOP) was positive and passed the significance test, indicating that the higher the population density, the higher the PLUI. The influence coefficient of the traffic factor (lnTRA) was positive and passed the significance test, indicating that the higher the density of public transport stations, the higher the PLUI.

The estimated results of the straightness centrality equation (Model 5) are shown in Table 4. The influence coefficient of PLUI (lnLUI) was positive and passed the significance test, indicating that an improvement in PLUI could promote an improvement in straightness centrality. The coefficient of the lag term (W × lnSC) of the straightness centrality was negative and passed the significance test, indicating that an increase in straightness centrality in the adjacent areas could lead to a decrease in local straightness centrality. The influence coefficient of the economic factors (lnEC) was positive and passed the significance test, indicating that improving the level of economic development was conducive to promoting an improvement in straightness centrality. The influence coefficient of the location factor (lnLOC) was negative and passed the significance test, indicating that the farther away from the city center, the lower the straightness centrality of road nodes.

The estimated results of the PLUI equation (Model 6) are shown in Table 4. The coefficient of the lag term (W × lnSC) of the straightness centrality was negative and passed the significance test, indicating that the straightness centrality in adjacent areas had a negative effect on the local PLUI. The influence coefficient of the straightness centrality (lnSC) was positive and passed the significance test, indicating that an improvement in straightness centrality can promote an improvement in PLUI. The coefficient of the lag term (W × lnLUI) of the PLUI was positive and passed the significance test, indicating that improving the PLUI in the adjacent areas could promote an improvement in the local PLUI. The influence coefficient of the economic factors (lnEC) was positive and passed the significance test, indicating that improving the level of economic development could promote an improvement in the PLUI. The influence coefficient of the location factor (lnLOC) was negative and passed the significance test, indicating that the farther away from the city center, the lower the PLUI. The influence coefficient of population size (lnPOP) was positive and passed the significance test, indicating that the greater the population density, the higher the PLUI.

## 5. Discussion

### 5.1. Impact of SC on LUI

The statistical results of the significant influence coefficient of SC on LUI are shown in Table 5. The influence coefficients of closeness centrality on CLUI, RLUI and PLUI were all positive, indicating that improving the closeness centrality was conducive to improving regional LUI, which is consistent with the research results of Yin et al. [28]. The impact of closeness centrality on LUI was as follows: public service land > residential land > commercial land. This may be attributed to the wide distribution of public service land in the main urban area of Jinan, which was more strongly influenced by the accessibility of the transportation network. The influence coefficients of betweenness centrality on CLUI, RLUI and PLUI were all negative, indicating that increasing the closeness centrality of the transportation network could lead to a decrease in regional LUI. The impact of betweenness centrality on LUI was as follows: public service land > residential land > commercial land. This may be attributed to the wide distribution of public service land in the main urban area of Jinan, which was more strongly influenced by the betweenness centrality of the transportation network. The influence coefficients of straightness centrality on CLUI, RLUI and PLUI were all positive, indicating that improving the straightness centrality of the transport network was conducive to improving regional LUI. The impact of straightness centrality on LUI was as follows: public service land > commercial land > residential land. This may be attributed to the fact that public service land and commercial land were more affected by the straightness centrality of the transportation network. In general, the SC in the main urban area of Jinan had the strongest effect on the PLUI and the least impact on the CLUI, which is contrary to the research results of some European cities [55,56]. This may be attributed to the wider distribution of public service land in the main urban area of Jinan. For example, public service facilities such as hospitals and schools are highly concentrated in the main urban area of Jinan.

The old urban area is the central area of the main urban area of Jinan, where commercial facilities are highly concentrated and land use intensity is high. This is different from the infrastructure distribution of major cities in the United States [57]. In the fringe area of the main urban area, owing to it being far away from the urban center, there are fewer commercial facilities, residential facilities and public services, and the intensity of land use is low. Therefore, we can improve the distribution pattern of transportation facilities and improve the closeness and straightness centrality of the edge areas, which are conducive to improving the LUI.

### 5.2. Impact of LUI on SC

The statistical results of the significant influence coefficient of LUI on SC are shown in Table 6. The influence coefficients of CLUI on closeness centrality and straightness centrality were both positive, while the influence coefficient on betweenness centrality was negative, indicating that improving CLUI was conducive to improving the accessibility and traffic efficiency of the transport network, but could lead to a decline in the mediating effect of the transport network, which is consistent with the research results of Chen et al. [45]. The impact of CLUI on SC was as follows: straightness centrality > closeness centrality > betweenness centrality. This may be attributed to the fact that the traffic efficiency of the traffic network in the main urban area of Jinan was more strongly influenced by the commercial land use intensity, which is similar to the study of Liu et al. [58]. The influence coefficients of RLUI on closeness centrality and straightness centrality were both positive, while the influence coefficient on betweenness centrality was negative. This shows that improving RLUI was conducive to improving the accessibility and traffic efficiency of the transportation network, but could lead to a decline in the intermediary role of the transportation network. The impact of RLUI on the SC was as follows: straightness centrality > betweenness centrality > closeness centrality. This may be attributed to the fact that the traffic efficiency of the transportation network in the main urban area of Jinan was more strongly influenced by residential land use. The influence coefficients of PLUI on closeness centrality and straightness centrality were both positive, while the influence coefficient on betweenness centrality was negative, indicating that improving the PLUI was conducive to improving the accessibility and traffic efficiency of the transport network, but could lead to a decline in the mediating effect of the transport network. The impact of the PLUI on the SC was as follows: betweenness centrality > straightness centrality > closeness centrality. This may be attributed to the fact that the mediating effect of the transportation network in the main urban area of Jinan was more strongly influenced by the PLUI. In general, the SC was most strongly influenced by the PLUI, which is consistent with the wide distribution of public service land in the main urban area of Jinan [28]. LUI had the strongest influence on the straightness centrality of the transportation network, which also shows that the traffic efficiency of the transportation network was most affected by LUI [59].

The traffic accessibility and traffic efficiency of the marginal areas of the main urban area of Jinan were low, meaning that they were not conducive to the sustainable development of urban traffic. Therefore, local government departments can improve the infrastructure construction of the region, such as building residential facilities and public service facilities of a certain scale, improving the intensity of regional land use, and thus improving regional traffic accessibility and traffic efficiency.

This study used the spatial three-stage least squares method to explore the two-way interaction between SC and LUI, which was innovative to a certain extent. However, this study also had the following shortcomings: First, the reclassified POI data were used to indirectly represent the urban functional land, but they were not scientific and accurate. In the analysis of influencing factors, owing to the difficulty in obtaining micro-data, this study selected fewer control variables, and did not consider the possible impacts of natural environmental factors and policy factors on LUI and SC.

The follow-up study will further expand the scope of study to multiple cities through a series of studies, and summarize whether there are general rules in different types of cities. Moreover, to overcome the difficulty of data acquisition, we will try to measure the commercial land, residential land and public service land more accurately based on plot area ratio, and compare these with the results of this study.

## 6. Conclusions and Policy Suggestions

This study explored the spatial distribution characteristics of SC and LUI in the main urban area of Jinan. Moreover, using the spatial three-stage least squares method, this study explored the two-way interaction between the SC and the LUI. The main research conclusions were as follows.

The closeness centrality showed an obvious “core-edge” pattern. Betweenness centrality and straightness centrality showed a multicentric structure. CLUI showed a multi-core distribution pattern in space. CBD was the largest agglomeration center, and several secondary agglomeration centers were distributed around it. RLUI and PLUI showed the characteristics of spatial distribution with the coexistence of large and small cores. 

There was an interactive relationship between SC and LUI. Closeness centrality and straightness centrality had positive effects on LUI, and LUI had a positive effect on closeness centrality and straightness centrality. Betweenness centrality had a negative effect on LUI, and LUI had a negative effect on betweenness centrality.

Moreover, the influence of the road network density and of the location factors on SC were the strongest. The greater the road network density and the closer to the city center, the higher the closeness centrality and straightness centrality, and the lower the betweenness centrality. Location factors, population density and traffic factors had the strongest effect on LUI. The closer the distance to the city center, the greater the population density, and the higher the density of bus stations, the higher the LUI.

According to the research results, this paper put forward the following policy suggestions for the future urban development of Jinan. First of all, the local government can formulate reasonable urban development policies, optimize the land use structure of the old urban area of Jinan, and promote the transformation and upgrading of transportation facilities in the old urban area. Secondly, it can improve the infrastructure construction of the new urban area and enhance the land use intensity of the new urban area. Finally, it can coordinate the construction of new and old urban areas, which is conducive to promoting the coordinated development of urban transportation and land use.

## Figures and Tables

**Figure 1 ijerph-20-05127-f001:**
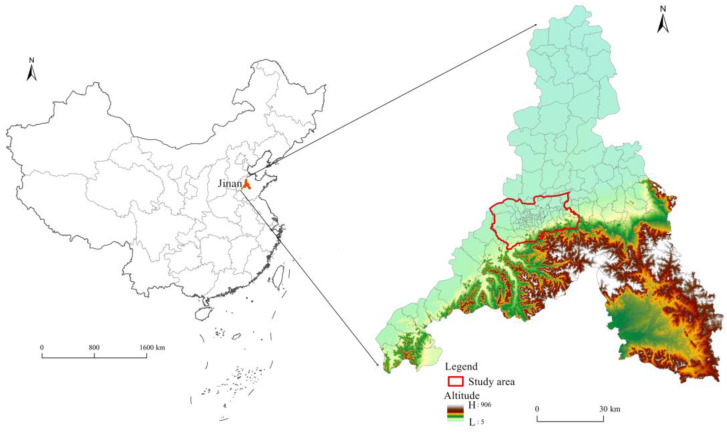
Study area.

**Figure 2 ijerph-20-05127-f002:**
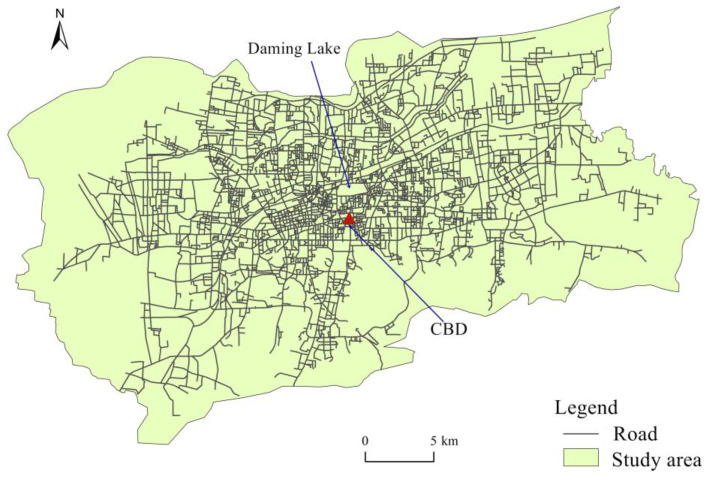
Distribution of road networks in the main urban area of Jinan.

**Figure 3 ijerph-20-05127-f003:**
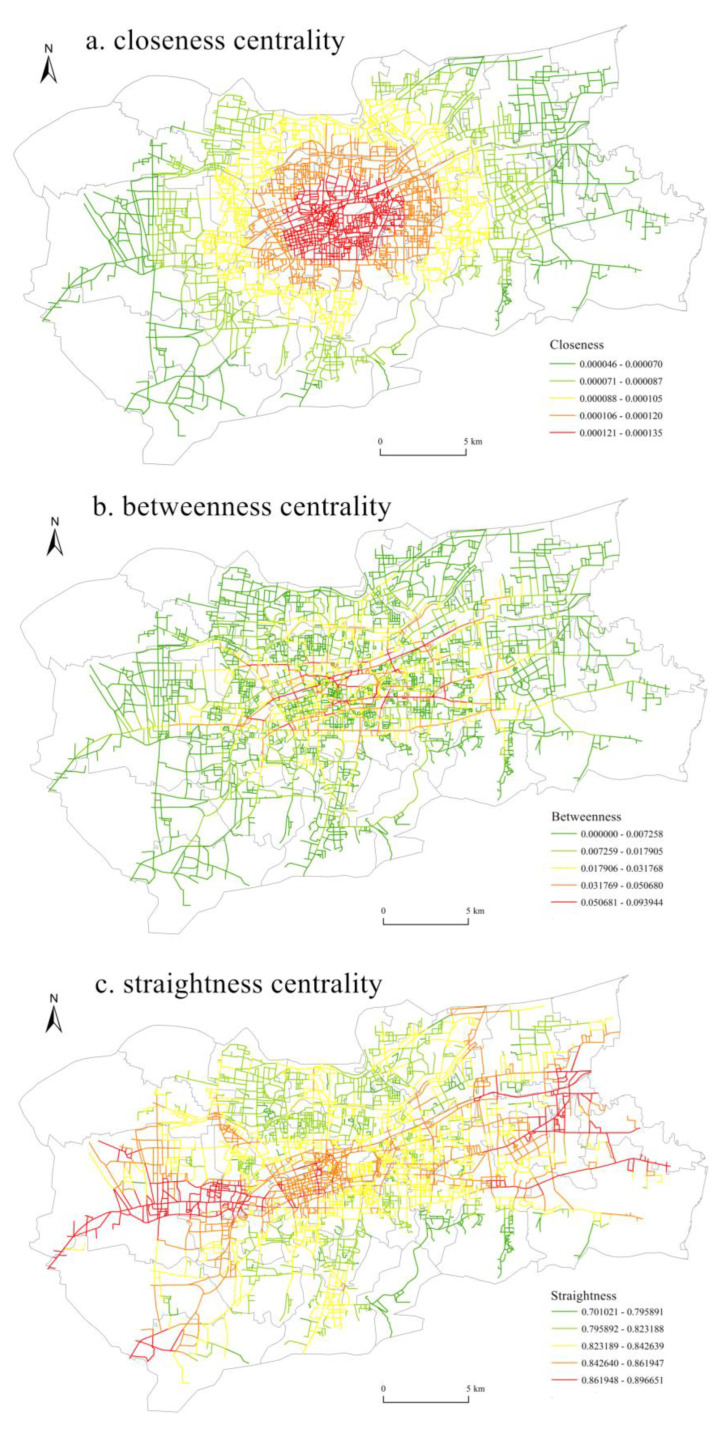
Spatial distribution of SC.

**Figure 4 ijerph-20-05127-f004:**
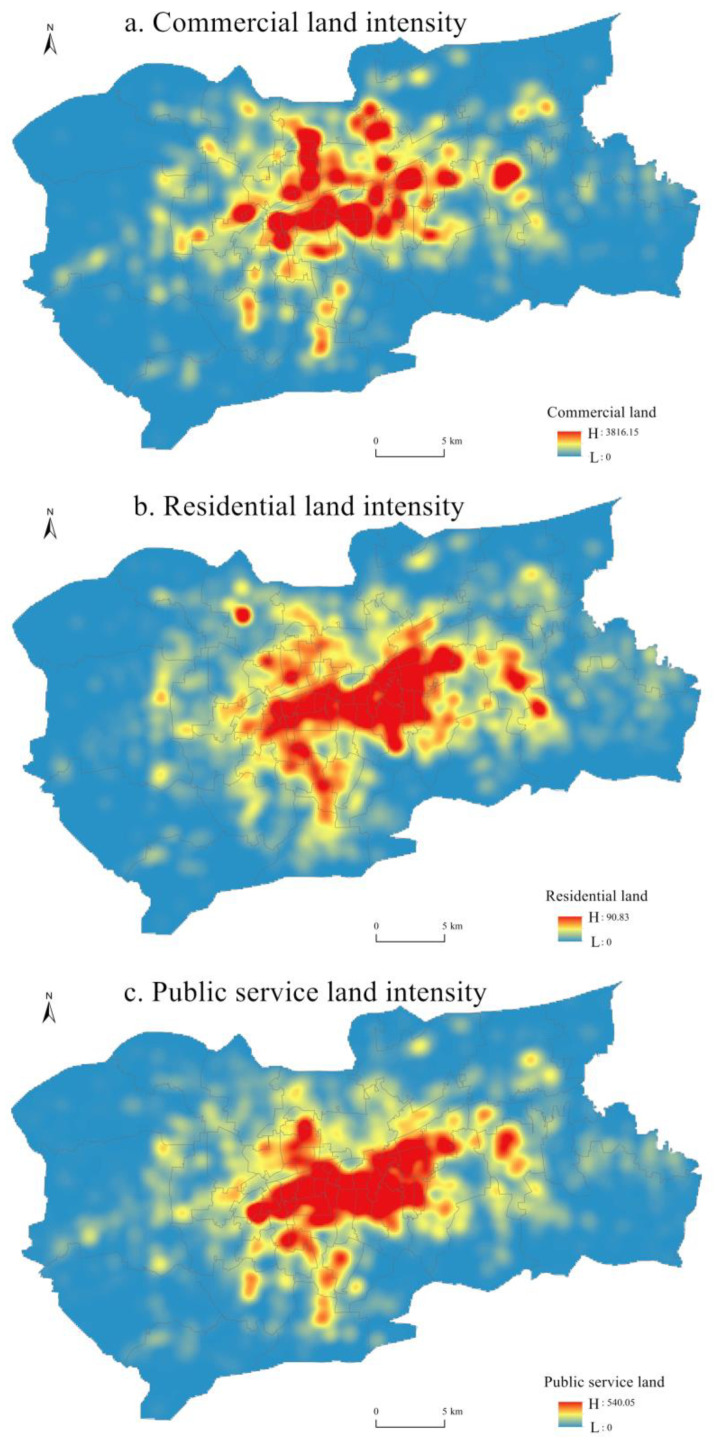
Spatial distribution of LUI.

**Table 1 ijerph-20-05127-t001:** Type and quantity of POI in the main urban area of Jinan.

Classification	Specific Categories	Number of POIs	Percentage (%)
Commercial facilities	Catering services	55,024	21.9
	Shopping services	85,708	34.0
	Financial and insurance services	4417	1.7
	Life services	42,957	17.1
	Accommodation services	6862	2.7
Residential facilities	Residential services	9657	3.8
Public service facilities	Communal facilities	2147	0.9
	Scientific, educational and cultural services	18,540	7.4
	Government agencies and social groups	9852	3.9
	Healthcare services	11,413	4.5
	Sports leisure services	5219	2.1

**Table 2 ijerph-20-05127-t002:** Regression results of SC and CLUI.

	Model 1	Model 2	Model 3	Model 4	Model 5	Model 6
	Closeness	CLUI	Betweenness	CLUI	Straightness	CLUI
*W* × *lnLUI*	0.0017 **	−0.0126	0.0025 **	−0.0147	0.0036 *	0.1209
*W* × *lnSC*	0.0298 *	0.0169 **	−0.0136 *	0.0684 *	−0.0473 ***	0.3659 **
*lnSC*		0.0485 **		−0.0138		0.1259 *
*lnLUI*	0.0025 *		−0.0019 **		0.0139 *	
*lnND*	0.1490 ***		−0.0257 *		0.0381	
*lnEC*	0.0136	0.0369 *	−0.0264	0.0024	−0.0097	0.0654 **
*lnLOC*	−0.0539 *	−0.2698 ***	0.0258	−0.1147 **	−0.0256 *	−0.0439 *
*lnPOP*		0.1894 *		0.2953 *		0.1169 ***
*lnTRA*		0.0569 *		0.0357 **		−0.0258
*CONS*	12.3901 *	−23.0145 **	9.0260 **	−12.269 **	16.2307 *	−29.1687 *
*Adjusted R* ^2^	0.5369	0.7598	0.5882	0.7123	0.6671	0.8109

Notes: ***, ** and * refer to the 1%, 5% and 10% significance levels, respectively.

**Table 3 ijerph-20-05127-t003:** Regression results of SC and RLUI.

	Model 1	Model 2	Model 3	Model 4	Model 5	Model 6
	Closeness	RLUI	Betweenness	RLUI	Straightness	RLUI
*W* × *lnLUI*	0.0126 *	−0.0307 **	0.0135 *	−0.2961	0.0985 *	0.2669 ***
*W* × *lnSC*	−0.0748	0.1189 **	−0.0473 *	0.1602	−0.0354 **	0.4690
*lnSC*		0.0721 ***		−0.1068 **		0.0367 *
*lnLUI*	0.0225 **		−0.0317 *		0.0453 **	
*lnND*	0.2490 ***		−0.1217 ***		0.0391	
*lnEC*	0.0736 *	−0.0464	−0.2271	0.0012	−0.0141	0.2474 *
*lnLOC*	−0.1079 **	−0.2608	0.0754	−0.1744 *	−0.0356 ***	−0.0149 **
*lnPOP*		0.4814 ***		0.2235 **		0.1769 *
*lnTRA*		0.0589 *		0.0827		−0.0754
*CONS*	11.3410 **	−27.4171 ***	10.1586 **	−17.3792 **	15.2487 **	−25.3645 **
*Adjusted R* ^2^	0.5524	0.6528	0.5779	0.6130	0.6472	0.7158

Notes: ***, ** and * refer to the 1%, 5% and 10% significance levels, respectively.

**Table 4 ijerph-20-05127-t004:** Regression results of SC and PLUI.

	Model 1	Model 2	Model 3	Model 4	Model 5	Model 6
	Closeness	PLUI	Betweenness	PLUI	Straightness	PLUI
*W* × *lnLUI*	0.0238 **	−0.0420 *	0.0274	0.0971 *	0.1055	0.1098 *
*W* × *lnSC*	−0.1001	0.2438 ***	−0.0234 **	0.1450	−0.1174 ***	−0.4628 *
*lnSC*		0.1057 *		−0.1148 **		0.1359 *
*lnLUI*	0.0425 **		−0.2304 *		0.1143 *	
*lnND*	0.2290 **		−0.1717 **		0.1321	
*lnEC*	0.0708 *	−0.0244	0.1270	0.0682 **	0.0321 *	0.2414 ***
*lnLOC*	−0.3524 ***	−0.2688 **	−0.0741	−0.1844 *	−0.0347 **	−0.0246 *
*lnPOP*		0.3024 **		0.2045 ***		0.1069 **
*lnTRA*		0.0547		0.0270 *		−0.0851
*CONS*	12.0270 *	−22.4010 **	11.1086 **	−20.3542 ***	12.2180 *	−26.3045 **
*Adjusted R* ^2^	0.5794	0.6548	0.5174	0.6670	0.6034	0.6998

Notes: ***, ** and * refer to the 1%, 5% and 10% significance levels, respectively.

**Table 5 ijerph-20-05127-t005:** Influence coefficient of SC on LUI.

	Com	Res	Pub
*Clo*	0.0485	0.0721	0.1057
*Bet*	−0.0138	−0.1068	−0.1148
*Str*	0.1259	0.0367	0.1359

**Table 6 ijerph-20-05127-t006:** Influence coefficient of LUI on SC.

	Clo	Bet	Str
*Com*	0.0025	−0.0019	0.0139
*Res*	0.0225	−0.0317	0.0453
*Pub*	0.0425	−0.2304	0.1143

## Data Availability

Not applicable.

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
