# Peer review of "The Interactive Relationship between Street Centrality and Land Use Intensity—A Case Study of Jinan, China"

_ijerph, 2023, doi:10.3390/ijerph20065127_

Round 1

Reviewer 1 Report

 It is need to making the following amendments and improvements.

 ï¼ˆ1)Commercial land intensity, Residential land intensity and Public service land intensity are suggested to be replaced by symbols. Such as Commercial land intensity can be replaced by CLUI.

(2)“3.2.3. Space three-stage least squares method” can be considered to be integrated with “4.2.1. Model establishment”.

(3)“4.2. Interactive relationship between SC and LUI” is suggested to be further integrated and refined.

(4) In order to enrich the content of the article. It is suggested to add policy suggestions for the sustainable development of Jinan in the article. 

(5)In order to facilitate the review of the article content. It is recommended to add page numbers in the text.

Author Response

Point 1:  Commercial land intensity, Residential land intensity and Public service land intensity are suggested to be replaced by symbols. Such as Commercial land intensity can be replaced by CLUI.

Response 1: Thank you for your suggestions. We have revised the full text according to your comments.

Point 2: “3.2.3. Space three-stage least squares method” can be considered to be integrated with “4.2.1. Model establishment”.

Response 2: Thank you for your comments. We have adjusted this part according to your comments.

Point 3: “4.2. Interactive relationship between SC and LUI” is suggested to be further integrated and refined.

Response 3: Thank you for your suggestions. We have deleted the redundant content in this section.

Point 4:  In order to enrich the content of the article. It is suggested to add policy suggestions for the sustainable development of Jinan in the article.

Response 4: Thank you for your suggestions. We have added policy suggestions at the end of the article.

Point 5: In order to facilitate the review of the article content. It is recommended to add page numbers in the text.

Response 5: Thank you for your comments. The journal editor has added page numbers to the article.

Reviewer 2 Report

Dear Authors,

It is a great job that you presented in the manuscript.

Discussion part is quite short, you must revise it and put relevant additional information and discussion with a wider range of previous publications.

Otherwise the manuscript has great scientific merit, I accept for publication after extending the discussion part.

Regards

Author Response

Point 1: Discussion part is quite short, you must revise it and put relevant additional information and discussion with a wider range of previous publications.

Response 1: Thank you for your comments. We have supplemented and improved the discussion part of the article.

Reviewer 3 Report

The study explored the interactive relationship between street centrality (SC) and land use intensity (LUI) in the main urban area of Jinan using spatial three-stage least square method. The study could provide a reference for guiding better urban sustainable development. I think the paper is valuable but have some concerns for the authors to consider:

1. To make the introduction a little more attractive the authors should be deliberate in stating how or where the study 'fits-in' with and add to what we know in the field. The innovativeness of the article needs to be highlighted.

2. Check for additional recent publications relevant to this paper on urban traffic and land use (i.e. 2020,2021,2022,2023). Reviews of existing research should be more summative.

3. The name of the lower-level administrative district of Jinan City should be clearly marked in the figure of study area.

4. The specific manifestations of Space three-stage least squares method should be explained.

5. The description of data sources and processing should be presented in a more organized manner.

6. 4.2.1. Model establishment should be moved to the section of 3.2.3. Space three-stage least squares method. The section of results should present the findings of the study.

7. The section 4.2. Interactive relationship between SC and LUI is too long and should be shortened for better reading and understanding. The summary of the findings of interactive relationship between SC and LUI should be presented in this section.

8. Contributions and comparisons of previous studies should be discussed as a focus and a separate chapter.

9. The English language should be checked and improved in several parts of the manuscript.

Author Response

Point 1: To make the introduction a little more attractive the authors should be deliberate in stating how or where the study 'fits-in' with and add to what we know in the field. The innovativeness of the article needs to be highlighted.

 Response 1: Thank you for your comments. According to your suggestion, we have added appropriate content in the introduction to enhance the attractiveness and innovation of the article. On page 2-3.

Point 2: Check for additional recent publications relevant to this paper on urban traffic and land use (i.e. 2020,2021,2022,2023). Reviews of existing research should be more summative.

Response 2: Thank you for your suggestions. We have summarized the existing research (on page 5) according to your suggestions, and added relevant literature published recently. The supplementary references are as follows:

  1. Shen, T.;Hong, Y.; Thompson, M. M.; Liu, J.; Huo, X.; Wu, L. How does parking availability interplay with the land use and affect traffic congestion in urban areas? The case study of Xi’an, China. Sustainable Cities and Society. 2020, 57, 102126.
  2. Zhou, H.;Gao, H. The impact of urban morphology on urban transportation mode: A case study of Tokyo. Case Studies on Transport Policy 
  3. Bao, Z.; Ou, Y.; Chen, S.; Wang, T. Land Use Impacts on Traffic Congestion Patterns: A Tale of a Northwestern Chinese City. Land 2022, 11, 2295.
  4. Sarzynski, A.; Wolman, H.L.; Galster, G.; Hanson, R. Testing the conventional wisdom about land use and traffic congestion: The more we sprawl, the less we move? Urban Stud. 2006, 43, 601–

Point 3: The name of the lower-level administrative district of Jinan City should be clearly marked in the figure of study area.

Response 3: Thank you for your comments. We determined the scope of the study area based on the distribution of the traffic network, rather than the boundary of the administrative region. Therefore, the name of the lower-level administrative region didn’t need to be shown in the figure of study area.

Point 4: The specific manifestations of Space three-stage least squares method should be explained.

Response 4: Thank you for your suggestions. We have explained the Spatial three-stage least squares method according to your suggestion. On page 7.

Point 5: The description of data sources and processing should be presented in a more organized manner.

Response 5: Thank you for your suggestions. We have adjusted the structure of data source and processing according to your requirements. We adjusted the content as follows:

The traffic network data of Jinan comes from the China National Geographic Information Public Service Platform. POI data can record the attribute information and spatial location of buildings or geographical entities. It covers a large number of data samples and contains a very rich amount of information. POI data of Jinan in 2020 was collected based on Gaode API.   

The ArcGIS software was used to extract the highways, main roads, secondary roads and main branches of Jinan main urban area. Secondly, disordered and disconnected routes were eliminated, and repeated routes were screened and combined with remote sensing satellite images. The traffic network data of the study area after processing was shown in Figure 2.

The POI data can intuitively reflect the spatial distribution of various urban functional elements and facilities. This paper used the re-classified POI data kernel density smoothing results to indirectly represent the LUI. Firstly, data cleansing was performed on the obtained POI data. Then, the required POI data of commercial facilities, residential facilities and public service facilities were selected for research (Table 1).

Point 6: 4.2.1. Model establishment should be moved to the section of 3.2.3. Space three-stage least squares method. The section of results should present the findings of the study.

Response 6: Thank you for your comments. We have adjusted this part according to your comments. On page 7-8.

Point 7: The section 4.2. Interactive relationship between SC and LUI is too long and should be shortened for better reading and understanding. The summary of the findings of interactive relationship between SC and LUI should be presented in this section.

Response 7: Thank you for your suggestions. We have deleted the redundant content in this section. We summarized the relationship between the two in the conclusions part of the article.

Point 8: Contributions and comparisons of previous studies should be discussed as a focus and a separate chapter.

Response 8: Thank you for your suggestions. We have supplemented and improved the previous research contributions and comparisons according to your suggestions. We adjusted the content as follows:

Compared with existing studies, this paper has the following advantages: First, this study can help local governments to coordinate the relationship between urban transportation and land use, which is of great significance to reasonably guide the construction of urban new areas and promote urban sustainable development. Second, this study makes up for the deficiencies of the current quantitative research on the interactive relationship between urban transportation and land use. Third, this study took the concentrated area of urban transportation and land development in China (the main urban area of Jinan) as the research object, it is representative and valuable.

   In general, current academic circles mainly focus on one-way impact research on urban traffic and land use, such as the impact of urban traffic on land use or the impact of land use on urban traffic [42,43]. In the correlation study of urban traffic and land use, Pearson correlation coefficient is mainly used to study the correlation between the two [44,45]. However, there are few researches on the interactive relationship between urban traffic and land use, and the existing researches mainly focus on qualitative analysis. Moreover, in the research of urban traffic network, Spatial syntax and Multiple centrality assessment model were mainly used for quantitative evaluation.

Point 9: The English language should be checked and improved in several parts of the manuscript.

Response 9: Thank you for your comments. We have checked and improved the English language of the article.

Reviewer 4 Report

This article seems to me to be very complete and well explained in all its phases.

Map legends look small at 100%.

From my point of view, the article can be published in the current format.

Author Response

Point 1: This article seems to me to be very complete and well explained in all its phases.

Response 1: Thank you for your affirmation of the article.

Point 2: Map legends look small at 100%.

Response 2: Thank you for your comments. We have enlarged and modified the map legends.

Reviewer 5 Report

Dear Authors,

some suggestions

1. use more references in discussion section

2. check the references in the manuscript according to the journal instuction.

Best regards

Molla Katerina

Author Response

Point 1: use more references in discussion section.

Response 1: Thank you for your suggestions. We have added some references in the discussion section. On page 22-24.

Point 2: check the references in the manuscript according to the journal instuction.

Response 2: Thank you for your comments. We have checked the references in the manuscript according to the journal instuction.
